# Assessment of Physicochemical and In Vivo Biological Properties of Polymeric Nanocapsules Based on Chitosan and Poly(*N*-vinyl pyrrolidone-*alt*-itaconic anhydride)

**DOI:** 10.3390/polym14091811

**Published:** 2022-04-28

**Authors:** Kheira Zanoune Dellali, Mohammed Dellali, Delia Mihaela Raţă, Anca Niculina Cadinoiu, Leonard Ionut Atanase, Marcel Popa, Mihaela-Claudia Spataru, Carmen Solcan

**Affiliations:** 1Faculty of Technology, University Hassiba Benbouali, BP 151, Chlef 02000, Algeria; zanounekheira@yahoo.fr (K.Z.D.); m.dellali@univ-chlef.dz (M.D.); 2Faculty of Medical Dentistry, Apollonia University of Iasi, Pacurari Street, No. 11, 700511 Iasi, Romania; anca.n.cadinoiu@univapollonia.ro; 3Academy of Romanian Scientists, Splaiul Independentei Street, No. 54, 050094 Bucharest, Romania; 4Public Health Departament, Faculty of Veterinary Medicine, Ion Ionescu de la Brad University of Life Sciences, Mihail Sadoveanu Alley, No. 8, 700489 Iasi, Romania; mspatarufmv@yahoo.com (M.-C.S.); carmensolcan@yahoo.com (C.S.)

**Keywords:** nanoparticles, natural and synthetic polymers, drug delivery systems, biocompatibility, in vivo tests

## Abstract

Drug delivery is an important field of nanomedicine, and its aim is to deliver specific active substances to a precise site of action in order to produce a desired pharmacological effect. In the present study nanocapsules were obtained by a process of interfacial condensation between chitosan (dissolved in the aqueous phase) and poly(*N*-vinyl pyrrolidone-*alt*-itaconic anhydride), a highly reactive copolymer capable of easily opening the anhydride ring under the action of amine groups of chitosan. The formed amide bonds led to the formation of a hydrogel membrane. The morphology of the obtained nanocapsules, their behavior in aqueous solution of physiological pH, and their ability to encapsulate and release a model drug can be modulated by the parameters of the synthesis process, such as the molar ratio between functional groups of polymers and the ratio of the phases in which the polymers are solubilized. Although a priori both polymers are biocompatible, this paper reports the results of a very detailed in vivo study conducted on experimental animals which have received the obtained nanocapsules by three administration routes—intraperitoneal, subcutaneous, and oral. The organs taken from the animals’ kidney, liver, spleen, and lung and analyzed histologically demonstrated the ability of nanocapsules to stimulate the monocytic macrophage system without producing inflammatory changes. Moreover, their in vivo behavior has been shown to depend not only on the route of administration but also on the interaction with the cells of the organs with which they come into contact. The results clearly argue the biocompatibility of nanocapsules and hence the possibility of their safe use in biomedical applications.

## 1. Introduction

Nanotechnology has gained considerable attention during last decades, and the development of different types of nanoparticles for biomedical applications is a growing field of research. Over the past twenty years, a significant number of nanoparticulate systems, composed of different materials including lipids, polymers, and inorganic materials, have been proposed in the biomedical field [1,2,3]. These nanocarriers, usually ranging from 1 to 1000 nm and suitable for the delivery of drugs, hormones, genes, nucleic acids, or imaging agents, have been designed in order to obtain improved specificity, drug targeting, and delivery efficiency, thus reaching a maximal therapeutic effect with minimal side effects [4].

Drug delivery is an important field of nanomedicine, and its aim is to deliver specific active substances to a precise site of action in order to produce a desired pharmacological effect. The development of a drug delivery system is influenced not only by the target site but also by the route of administration and the nature of the nanocarriers [5].

Polymer-based vehicles constitute the main branch of drug delivery systems and can be used for both diagnostic and therapeutic purposes. Polymer carriers with a spherical shape and smooth texture are considered ideal for delivering chemotherapeutic agents in order to easily transport them through the vascular system [6]. The most-investigated polymer carriers are micelles, nanospheres, nanocapsules, dendrimers, and polymersomes [7,8,9,10]. Generally, all these carriers can increase the local concentration of loaded active substances and improve their delivery, especially when they are poorly water soluble or when their bioavailability is low.

At the present, the majority of studies regarding polymeric drug delivery systems are focused on optimizing the nanocarrier physicochemical parameters, such as size, physical stability, and drug loading efficacy, but also on carrying on preliminary in vitro cytotoxicity tests in order to prove the effectiveness of the obtained formulations [11]. However, in vivo tests, which are useful in the investigation of the biological effects of these polymeric nanomaterials, are often not taken into account as they can be difficult to carry out. Despite the remarkable rapidity of development of nanomedicine, relatively little is known about the interaction at the nanoscale of polymeric carriers with living systems. The behavior of nanocarriers in the body depends not only on the administration route but also on their interactions with the cells with which they come into contact.

Thus, in the present study, a series of original polymeric nanocapsules (NCs) based on chitosan (CS) and poly(*N*-vinyl pyrrolidone-*alt*-itaconic anhydride) (NVPAI), were firstly investigated from a physicochemical point of view, and then their biological features were determined by in vivo testing in order to demonstrate their effectiveness as safe drug delivery systems. At this point it is worth mentioning that the nanocapsules (NCs), due to their morphology, have some practical advantages with respect to other nanocarriers, as they can be characterized by increased drug encapsulation efficiency and thus by an enhanced therapeutic effect. According to its polarity, the active substance is incorporated in the core, which then acts as a reservoir, or possibly adsorbed or covalently attached to the polymeric shell [12].

These hollow NCs were obtained by interfacial condensation method in absence of any kind of toxic crosslinking agents and in normal conditions (temperature and pressure). The polymer shell was formed by amide bridges at the contact between the NH_2_ groups of CS and highly reactive anhydride cycles of poly(NVPAI). It has to be mentioned that a similar type of NCs was already prepared and characterized by our research group [13], but the novelty of the present system is that the exterior layer of the polymeric shell of the NCs is formed by the synthetic NVPAI copolymer.

CS was used in this study as it is one of the biopolymers that can form nanoparticles with unique properties and therefore is currently receiving great interest for drug delivery and tissue engineering applications in the medical and pharmaceutical field due to its interesting features, such as biocompatibility, biodegradability, and non-toxicity [14]. Compared with natural polymers, synthetic polymers such as NVPAI have high purity and good reproducibility and can control the release time of loaded active substances [15].

The obtained NCs have been analyzed structurally by Fourier transform–infrared spectroscopy (FT-IR), morphologically by TEM, and gravimetrically by thermo-gravimetric analysis (TGA). Moreover, the resulting NCs were characterized in terms of particle sizes, zeta potential, drug encapsulation efficiency, and in vitro drug release kinetics by using a hydrophilic model drug, 5-fluorouracil (5-FU). Furthermore, in vivo tests were performed on adult albino BALB/c line mice using different concentrations of NC suspensions and three types of administration routes, such as intraperitoneally, subcutaneously, and per o.s. for 21 days.

## 2. Experimental Part (Materials and Methods)

### 2.1. Materials

Chitosan (CS) (low molecular weight, degree of deacetylation 91%), acetone, dimethyl sulfoxide (DMSO), hexane, surfactants (Tween 80, Span 80), and drug (5-Fluorouracil), were purchased from Sigma Aldrich (St. Louis, MO, USA). Poly(N-vinylpyrrolidone-alt-itaconic anhydride) (NVPAI) is an alternant copolymer synthesized in laboratory by a radical copolymerization method [13]. Also, phosphate-buffered solution (PBS) with pH = 7.4 and double-distilled water was prepared in our laboratory.

### 2.2. Preparation Method of Nanocapsules

CS/poly(NVPAI)-based nanocapsules (NCs) were obtained by the interfacial condensation method. Initially, the polymer solutions were prepared in different phases. The aqueous phase was obtained by dissolving a specific amount of CS (Table 1) in 20 mL of 2% acetic acid solution (0.4 mL acetic acid was added to 20 mL distilled water) at a temperature of 65 °C under magnetic stirring. The solution was filtered before use and brought to room temperature. Then, an appropriate quantity of non-ionic surfactant Tween 80 (2% *w*/*v*) was added in the polymer solution and well homogenized. Separately, the organic phase was prepared by dissolving exactly 500 mg of poly(NVPAI) in 15 mL of DMSO under magnetic stirring. After complete dissolution of the copolymer, a specific volume of acetone was added, according to Table 1, under continuous stirring followed by the addition of the hydrophobic surfactant Span 80 (2% *w*/*v*). After this dissolution step, the aqueous solution of CS was slowly added drop-wise into the organic solution of poly(NVPAI) under vigorous magnetic stirring at room temperature. After 2 h, the formed NCs were separated from supernatant by centrifugation for 20 min at 7500 rpm. NCs were then purified by successive washes with distilled water (7 times) and acetone (5 times). After the last wash with acetone, the product was washed twice with hexane. Finally, the obtained product was dried from hexane at room temperature to constant weight.

The variables taken into account in this study were the molar ratio between the functional groups involved in the condensation reaction, respectively NH_2_/anhydride cycles, expressed in the Table 1 as the ratio between the two polymers [CS/poly (NVPAI)], and the volume ratio between the aqueous and organic phases.

The yield of the NCs (Table 2) was calculated according to the following equation:(1)$$\mathrm{NCs}\;\mathrm{yield}\;\left(\%\;\right)=\frac{amount\;of\;recovered\;nanocapsules}{total\;amount\;of\;polymers\;used}\;*\;100$$

### 2.3. Characterization Methods

#### 2.3.1. Structural Characterization

The structural characterization of NCs was accomplished spectrally by Fourier transform infrared spectroscopy (Schimadzu Corporation, Kyoto, Japan) (FTIR) to confirm the formation of new amide groups. The structural characterization was performed with a IRSpirit FTIR Spectrometer spectrometer. All samples were prepared as KBr pellets and scanned over the wave number range of 400–4000 cm^−1^ at a resolution of 4.0 cm^−1^. The relevant bands in the absorption spectrum have been attributed to corresponding functional groups.

#### 2.3.2. Size, Morphology and Zeta Potential of NCs

Transmission electron microscopy (TEM) was used to determine the size, shape, and surface morphology of NCs. The samples for transmission electron microscopy (TEM) were prepared by slow evaporation of a suspension in acetone on a formvar-coated copper grid. The samples were analyzed with a Philips CM100 microscope equipped with an Olympus camera and transferred to a computer equipped with the Megaview system.

The mean diameter of NCs in dispersion and the size distribution were determined in triplicate at 25 °C at a suspension concentration of 1% (*w*/*v*) by dynamic light scattering (DLS) (Zeta Nanosizer Malvern). Anhydrous acetone was used as a dispersant to evaluate the average diameter of unswollen NCs. The particles’ diameter in physiological saline solution was also evaluated, this medium being similar to the in vivo medium. This evaluation was performed as soon as possible after NCs came into contact with the aqueous environment. The zeta potential of NCs was determined by electrophoresis in phosphate buffer solution (PBS; pH = 7.4).

#### 2.3.3. Thermal Properties

Thermogravimetric analyses (TGA) have allowed the determination of sample weight loss as a function of temperature. These analyses were accomplished with a TA Instrument Q600 analyzer in air atmosphere (100 mL/min) with a heating rate of 10 °C/min, in a temperature range from the room temperature to 700 °C. The nanocapsules samples with a weight between 8–10 mg were heated in a platinum crucible. The operation parameters were kept constant for all the tested samples. The thermal analysis results were processed with the Universal Analysis (V 2.0) software (TA instruments, New Castle, DE, USA).

#### 2.3.4. Swelling Behaviour in Aqueous Solutions

In order to predict and understand the behaviour of NCs during the encapsulating and release process, and therefore to assess their behaviour as potential drug carriers, swelling studies were performed by gravimetric method. The swelling degree of the NCs samples was analyzed in slightly alkaline aqueous medium, pH = 7.4, that simulates physiological conditions. A specific amount (0.03 g) of dried NCs were weighted and immersed in Eppendorf’s tube containing PBS. The obtained suspension was maintained at 37 ± 0.5 °C under magnetic stirring at 120 rpm. At pre-set times, the suspension was centrifuged, the supernatant was removed, and the swollen sample was weighed. All experiments were performed in triplicate. To ensure the swelling until equilibrium, the samples were allowed to swell for 24 h. The percentage of swelling ratio (*Q*%) was determined with Equation (2)
(2)$$Q\left(\%\right)=\frac{W-W_{0}}{W_{0}}*100\;$$
where *W* is the weight of swollen sample (mg) and *W*_0_ is the initial weight of dry sample (mg).

In parallel with the gravimetric method, the change in the size of the NCs on swelling was also evaluated.

#### 2.3.5. Drug Encapsulating Studies

The drug encapsulating process was carried out through diffusional mechanism. In this study, 5-Flourouracil (5-FU) was used as the model drug. Briefly, 20 mg NCs were dispersed in 1.5 mL aqueous drug solution with a concentration of 10 mg 5-FU/mL in ultrapure water. The suspension was maintained under magnetic stirring (120 rpm) and temperature (37 °C) for 24 h. The drug-loaded NCs were separated from supernatant by ultracentrifugation at 8000 rpm for 10 min. The drug-loaded NCs were lyophilized and stored as powder for further analyses. The amount of 5-FU encapsulated into NCs was calculated by the difference between the initial amount of 5-FU in solution and the amount of 5-FU from supernatant using a UV Spectrometer (Nanodrop One, Thermo Scientific, Waltham, MA, USA) at 266 nm [13,16]. The encapsulation efficiency (*E_ef_*%) of 5-FU into NCs was calculated as follows:*m_l_* = *m_i_* − *m_s_*
(3)
(4)$$E_{ef}\%=\frac{m_{i}-m_{s}}{m_{i}}\times 100$$
where *m_l_*—the amount of encapsulated 5-FU (mg); *m_i_*—the initial amount of 5-FU (mg); *m_s_*—the amount of 5-FU found in supernatant (mg).

The obtained drug-loaded NCs are designated as follows: CN-1-5FU, CN-2-5FU, CN-3-5FU, CN-4-5FU, CN-5-5FU, CN-6-5FU, and CN-7-5FU.

#### 2.3.6. In Vitro Drug Release

The in vitro drug release studies were realized by the dialysis method. Each sample of 5-FU-encapsulated NCs was introduced into a dialysis membrane and, after that, was individually immersed in flasks with 13 mL PBS at pH = 7.4, a value which is similar to blood. This system was maintained at 37 ± 0.5 °C under continuous stirring at 120 rpm for all of the release period. At regular time intervals, 1 mL of solution was taken and replaced with fresh PBS. The 5-FU released and present in the medium was spectrophotometrically determined at 266 nm wavelength, using a Nanodrop One (Thermo Scientific). The release efficiency of 5-FU (*R_ef_*%) was calculated using Equation (5):(5)$$Ref\left(\%\right)=\frac{m_{r}}{m_{l}}\times 100\;$$
where *m_r_*—the amount of drug released from NCs (mg); *m_l_*—the amount of the drug encapsulated into the NCs (mg).

#### 2.3.7. In Vivo Testing

The study was conducted on 42 adult albino BALB/c line mice, aged approximately 4 months, reared under conventional laboratory conditions (20–23 °C, 55% UR), fed standardized pelleted feed, fruits and vegetables, and water at discretion. Their bodyweight was monitored in the first, tenth, and last day of the experiment, and it was constant within the experimental error limits (Appendix A). Mice undergoing the experiment (equally female and male) were divided into 7 groups: one control and 6 experimental groups of 6 mice. To each group, CN-4 and CN-6 suspensions were administered daily by three different routes, such as intraperitoneally (0.01 mL and 0.02 mL), subcutaneously (0.1 and 0.2 mL), and per o.s. (0.2 and 0.3 mL) for 21 days. Suspensions of CN-4 and CN-6 were obtained by adding 5 mL of saline solution over 6.25 mg powder, the suspension being prepared approximately 2 h before use. The health status of the mice and body weight dynamics were followed and 2 days after the completion of the experiment the mice were euthanized by cervical dislocation and probes were harvested from internal organs (kidney, liver, spleen, and lung) and processed for histological examination.

#### 2.3.8. Histological and Immunohistological Analysis

Organs samples were fixed in 10% formalin solution for 24 h. Approximately 0.5 cm-thick slices were dehydrated with a decreasing concentration of ethanol solution, then clarified in xylene and embedded in paraffin. After being cut with the microtometer, 10 microscope slides from each paraffin block were selected, specifically stained, and read on the Olympus CX41 microscope. They were initially stained with hematoxylin eosin (HE) then IHC using 4 antibodies: Cd147, p65, alpha SMA, HMC II, and Cox-2. Anti p65 (Nuclear factor-kB p65) antibodies, AA 143–158, antibodies, CD147 (ab188190), α-SMA (anti-alpha smooth muscle actin antibody), MA5-11547 (14A-asm-1), MHC II (Dako M0746), and Cox-2 (ab16701 SP-21), were used to perform immunohistochemical staining. After sections were deparaffinized in Xylen, hydrated in ethanol, and microwaved for 10 min at 95 °C in 10 mmol citrate acid buffer pH6, they were cooled for 20 min, then washed twice in PBS for 5 min. Slices were treated with 3% hydrogen peroxide and rinsed with PBS, after which they were incubated overnight at 4 °C in a humid atmosphere with primary antibodies in dilutions of 1:100 CD147, MHC II, Cox-2 and 1:500 p65, α-SMA. The following day, slides were washed 3 times in PBS for 5 min, being incubated with secondary antibodies. For CD147, MHC II, and Cox-2 activity of bone cells, goat anti-rabbit IgG secondary antibody was used, and for p65 and α-SMA, goat anti-mouse IgG secondary antibody was chosen. Microscope slides were developed in 3,3′-diaminobenzidine (DAB) and finally counterstained with hematoxylin. Images were interpreted using ImageJ IHC profile software.

DAB IHC profile scores were negative (−), low positive (±), positive (+), and over positive (++). Scoring of HE histological lesions was done by assessing changes and scoring as follows: no change (−), minor (+), medium (++), and major (+++).

### 2.4. Statistical Analysis

The statistical significance of cytotoxic activity was analyzed by Student’s *t*-test. The values are expressed as mean ± SE of three parallel measurements, *p* < 0.05 being considered significant.

## 3. Results and Discussion

The objective of the present study was to assess the physicochemical and biological properties of a series of NCs based on CS and poly(NVPAI) which can be further used as drug delivery systems for the controlled and sustained release of different types of drugs.

A first result is that the yield of obtaining NCs increases with increasing the amount of CS in their composition (respectively of the CS/poly(NVPAI) molar ratio), as can be noticed in Table 2.

The increase in the initial amount of chitosan leads to the increase of the bonds formed between the poly(NVPAI)- and CS-reactive groups because a higher number of these reactive groups are available and can be involved in the interfacial polycondensation reaction. Consequently, a smaller amount of non-crosslinked polymer chains remains in the system and can be eliminated during the purification and washing process of the nanocapsules. In addition, enhancement of the aqueous phase/organic phase ratio, at the same CS/poly(NVPAI) ratio, has led to a visible increase in the nanocapsules final yield [17].

Different analysis techniques were used, and the obtained results are presented in the following.

### 3.1. FTIR Spectroscopy

The FT-IR spectra of CS, poly (NVPAI), and NCs presented in Figure 1 confirmed the reaction between the anhydride groups of the copolymer and chitosan amine groups. Absorption bands at approximately 1778 cm^−1^ and 1858 cm^−1^ from the copolymer spectrum correspond to the stretching vibration of the –C=O anhydride groups. Another important peak is located at approximately 989 cm^−1^ and can be assigned to the C–O–C bond in the anhydride group [13].

CS sample presents a peak at 1652 cm^−1^ that corresponds to carbonyl groups (C=O) and another peak at 1601 cm^−1^ attributed to free amino groups (–NH2).

The FT-IR spectra of NCs samples evidenced the disappearance of characteristic peaks for the anhydride groups and the appearance of the absorption bands at 1647 cm^−1^ and 1658 cm^−1^ corresponding to a carbonyl bond of the newly formed amide groups. The appearance of absorption bands at 1715 cm^−1^ (for sample CN-6) and at 1721 cm^−1^ (for sample CN-4) that are attributed to the –C=O of the carboxylic groups reveals that some of the anhydride groups have hydrolyzed. Finally, the disappearance of absorption bands at 989 cm^−1^ evidenced that all anhydride groups participated in the amidation or hydrolysis reaction.

Figure 2 shows the FTIR spectra for simple 5-FU, CN-6 without drug and drug-loaded NCs (CN-4-5FU and CN-6-5FU). The peaks characteristic of 5-FU spectrum (1429.92 cm^−1^, 1246.89 cm^−1^, 812.19 cm^−1^, 755.00 cm^−1^, and 640.6 cm^−1^) are also found in the NC samples loaded with the model drug. In addition, these peaks are not visible in the spectrum of samples without drug (Figure 1 and Figure 2).

### 3.2. Size, Morphology and Zeta Potential of NCs

The mean diameters for NCs in acetone varied between 107 and 250 nm (Figure 3 and Table 2). Size distribution curves evidenced the presence of two populations of nanocapsules in the case of CN-1, CN-2, and CN-3 samples. CN-4, CN-5, CN-6, and CN-7 samples presented a monomodal distribution. From the obtained results, it can be noticed that the NCs’ diameter increases with the increasing of the quantity of CS. The explanation of this behavior is based on the fact that, by increasing the amount of CS, increasing amounts of polysaccharide are involved in the formation of the NCs’ membrane, which becomes thicker, thus contributing to the increase of the diameter.

Increasing the volume of the organic phase, the number of macromolecules of copolymer that come into contact with a drop of the aqueous solution of CS is decreased gradually due to the dilution of the copolymer solution, and therefore the diameter of NCs decreases.

The NCs swell very quickly in aqueous environments. After 5 min in the physiological saline solution, diameters between 1089 and 2256 nm were recorded. The amount of CS and the volume of the organic phase had the same influence on the NCs size, even in an aqueous environment.

The zeta potential values evidenced the stability of the aqueous dispersion of NCs in a medium with slightly alkaline pH. In Table 3 the zeta potential values are presented which varied between −8 mV and −23.21 mV. Negative charge of NCs can be attributed to the presence of carboxylic groups formed as a result of the interfacial condensation reaction by amidation of the anhydride cycle in poly (NVPAI) or by their hydrolysis. By condensation, a carboxylic group also appears in the reaction of an amine cleavage with the copolymer. By hydrolysis, each anhydride cycle generates two carboxylic groups. This explains why by increasing the amount of CS, the zeta potential is reduced: the explanation lies in the overall formation of fewer carboxylic groups. From Table 3 it can be seen that, in the case of NCs without 5-FU, the increasing of the CS amount in the system leads to a decrease in the zeta potential values. On the other hand, NCs loaded with 5-FU show an increase in the zeta potential value as the amount of CS in the composition of the NCs increases.

From the Figure 4 it appears that the NCs have a spherical morphology and a moderate polydispersity, but that their size is smaller than that in acetone obtained by DLS. This can probably be explained by the different principle of operation of the two techniques (in liquid media and in dry state). The solvent used for DLS, acetone, may diffuse to some extent inside the NCs causing a slight increase in its volume.

### 3.3. Thermal Behaviour

The evaluation of the temperature behavior of the NCs was performed, on one hand, in order to have additional evidence that their membrane is made of both polymers, and on the other hand, to determine whether it is possible to heat sterilize them before administration without suffering changes caused by possible thermal degradation. In Figure 5 are presented the TGA chromatograms of different NC samples.

From the TGA results provided in Figure 5, it is obvious that the obtained NCs are stable until 400 °C, when a second degradation step occurs. This is a proof that these NCs are stable during the sterilization step. Moreover, analysis of the TGA chromatograms showed that the composition of the NCs shell does not qualitatively influence its thermal degradation behavior. It is found that indeed the CN-6 sample (with a higher yield, previously explained by the higher amount of reacted copolymer) has a slightly higher calcination residue content compared to the CN-2 sample.

### 3.4. Swelling Degree of NCs

The swelling behaviour of the obtained NCs was investigated in slightly alkaline medium (pH 7.4). This medium was chosen as these drug delivery systems are intended to be used in biological fluids, and the obtained results are illustrated in Figure 6.

It is evident that all the NCs presented a high swelling degree which varies from 1476% to 1851%. The size, the composition, and the preparation parameters of NCs have an important influence on the swelling properties. In Figure 6a it can be observed that the swelling degree varied between 1851% and 1476% and decreased with the increase of the molar ratio of CS/NVPAI. The highest swelling degree was obtained for the CN-1 sample which is characterized by the smallest CS amount. The swelling of the NCs is caused by the water penetration within the empty core until complete filling as well as by the swelling of the polymer membrane, which has a hydrogel character [17].

The increase in the number of moles of -NH_2_ groups from CS has as a consequence the increasing of the crosslinking density of the network of which the capsule membrane is formed, which reduces the diffusion of the water inside the capsule but also the amount of water that swells the membrane. In the other hand, it can be observed that the swelling degree of the NCs increases with the increase of the volume organic phase (Figure 6b) for an identical CS/poly(NVPAI) molar ratio of 0.3. This behaviour is explained as follows: as the volume of the organic phase increases (dilution of the copolymer solution), the number of macromolecules of poly(NVPAI) which come into contact with CS at the interface of the aqueous solution with organic phase is decreased and becomes smaller and smaller, so the crosslinking density of the NCs membrane will decrease, and therefore, the swelling rate will increase [18].

Since NCs can be used in physiological saline solutions for medical applications, it has also been necessary to evaluate their size in this environment (Figure 7). For the CN-4 and CN-6 samples, which were used for in vivo tests, the influence of the aqueous medium on the size over time was also evaluated (Figure 8).

After 24 h the diameter of the NCs had values between 5700 and 10,300 nm. Being capsules, it is easy to understand why they grow so much in size in the aqueous environment. Aqueous solution penetrates very easily inside the capsule and the network that creates the membrane of the capsule can relax freely.

Figure 8 shows that the diameter of the two samples increases quite rapidly after they are introduced into the saline solution. After only 2 min they go from nanometers to micrometers, and after half an hour they almost reach equilibrium.

### 3.5. Encapsulation Efficiency of a Model Drug

This type of NC was developed in order to be used as efficient drug delivery systems for hydrophilic drugs. In this study, 5-FU was used as a model drug. The efficiency of encapsulating 5-FU into NCs and the amount of encapsulated drug (5-FU g/g NCs) are presented in Table 4.

### 3.6. Drug Release Kinetics

The release studies of the model drug, 5-FU, were performed in slightly alkaline medium (pH = 7.4) and the results are reported in Figure 9.

The results showed that 5-FU release from NCs was between 51% and 60%, and simple 5-FU was 100% released within 270 min of the start of the experiment. From Figure 9, it appears also that in the case of NCs loaded with 5-FU, a faster release was observed in the first hour which is due to the release of the drug adsorbed at the surface of NCs followed by a slow release of the encapsulated drug until the equilibrium was reached. Although the differences in release kinetics between the obtained NCs samples are not noticeable, there is still an influence on the amount of CS. If the CS amount increases, the reticulation density increases also, leading thus to a decrease of the drug release rate through the NCs shell. These results are in full accordance with the evolution of the previously discussed swelling process. These tests indicate that the obtained NCs can be used for the controlled and sustained release of hydrophilic drugs.

### 3.7. In Vivo Testing

For in vivo testing, mice were divided into 7 groups, one control and 6 experimental groups that received “CN” which received the NCs. At this point, different administration routes of the NCs were investigated.

Under light microscopy, the liver in the control group is normal; the hepatocytes are large and cuboidal with a prominent round nucleus and eosinophilic cytoplasm. The cords are radially arranged from the centrilobular venule to the periphery of the lobule. Hepatic sinusoidal capillaries are arranged between the hepatic plates with a sparse arrangement of Kupffer cells. In the control group, the kidney has a normal appearance of both the cortex and medulla. Malpighian corpuscles and urinary tubules show no changes (Figure 10). Hematoxylin- and eosin (HE)-stained sections from the control group (group I) showed normal histological architecture of the lungs: thin-walled alveoli, alveolar sacs, clear alveolar spaces, and thin-walled blood vessels. The epithelial lining of the alveoli was composed of squamous alveolar cells with dense nuclei (type I pneumocytes) and large alveolar cells with large, rounded nuclei (type II pneumocytes). The spleen in the control group is covered by a thin capsule of connective tissue and shows white pulp and red pulp without changes. In the liver, kidney, and lung, congestion of blood vessels was observed.

In the case of the experimental groups, by HE staining, no major changes in the histological structure of the organs studied were observed (Table 5).

CN-4 and CN-6 produced reduced histological changes in the organelles studied depending on the route of administration and dose. Administration p.o. and s.c., irrespective of dose, caused ectasia of a small number of veins in the liver (Figure 11), kidney (Figure 12), lung (Figure 13), and spleen (Figure 14). In the liver the hepatocytes show microvesicles and an increase in the number of Kupffer cells (6–7.5/12,879.8 mm^2^). In the lung there is a high number of alveolar macrophages residing in the septal wall (14–24/12,879.8 mm^2^). The alveoli have an intact, thin septal wall, but in some places it is thickened by macrophages. In the kidneys, the proximal convoluted tubules show rare brush border changes, and a small number of Malpighian corpuscles have a condensed glomerulus and increased capsular space volume (one in each preparation examined). In the interstitial space of the urinary tubules in the cortical area macrophages are observed (2–5/12,879.8 mm^2^). In the spleen the lymphoid follicles show a slight increase, where pigments and megakaryocytes, some of them giant, appear.

Administration of i.p. CN-4 0.01 mL and 0.02 mL in the liver, in hepatocytes with micro- and macrovesicles, resulted in 7–9.3 Kupffer cells /12,878.2 mm^2^, alveolar macrophages (9–13/12,879.8 mm^2^), and interstitial macrophages between the urinary tubules 4–6.6/12,879.8 mm^2^. A small number (1–3 out of 10 fields examined of each individual) of Malpighian corpuscles were found with sclerotic changes of the glomerulus, with dilated convoluted tubules and with damaged brush border. Administration of i.p. CN-6 0.01 mL and 0.02 mL caused ectasia of some veins in the liver, kidney, and lung. The 0.02 mL dose produced changes in hepatocytes represented by macrovesicles, small focal necrosis, frequent Kupffer cells (9–11/12,879.8 mm^2^), rare Malpighian corpuscles with sclerosing glomerular lesions (3 in each preparation examined), a reduced frequency of dilated proximal convoluted tubules with secretion in the lumen, macrophages in the interstitial space between the urinary tubules (5–7.5/12,879.8 mm^2^), and alveolar macrophages in the septal wall of the pulmonary alveoli (9–18.5/12,879.8 mm^2^). In the lung, clusters of 15–20 macrophages were observed in the lamina propria of the bronchioles and perivascular.

From this figure it can be seen that in all experimental groups, the main change is in congestive blood vessels and a higher frequency of Kupffer cells. The arrangement of hepatocytes in radial cords is preserved. Cell shape and nuclei are similar to the cells in the control group. Hepatocytes from experimental groups receiving CN-4 and CN-6 p.o. have hepatocytes with intracytoplasmic microvesicles. In the groups administered s.c. and i.p., cytoplasmic micro and macrovesicles and focal necrosis are observed.

From this figure it can be seen that in all experimental groups the main change is in the vascular congestion. In the experimental groups that received CN-4 and CN-6 p.o., a reduced number of interstitial macrophages appear among the urinary tubules in the cortical area. In the groups given s.c. and i.p., the Malpighian corpuscles with glomerular compaction, modified brush border in some proximal convoluted tubules, and small hemorrhages in the cortical area are observed. Some tubules with increased nephrocytes in volume were apparent.

Sections from the experimental groups showed preserved lung architecture. CN administration produced at some sites a change in alveolar septal diameter due to the accumulation of alveolar macrophages. Alveoli in the vicinity of congested vessels were observed in collapse. The number of alveolar macrophages located in the angular septal wall between alveoli was higher upon p.o. and i.p. administration. In the i.p. administration groups, macrophage clusters were observed in the lamina propria of the bronchi or perivascular. No alveolar macrophages were observed in the alveolar lumen.

In the experimental groups, the number of lymphoid follicles is increased, especially when administered i.p.; lymphatic cords have a slightly increased volume, and the number and frequency of megakaryocytes and pigment cells is higher in the experimental groups compared to the control group.

Following the evaluation of the IHC images, a low positive CD-4 label was observed for all markers regardless of the route of administration. Positive labeling was recorded for CN-6 only at intraperitoneal administration.

The CD147 marker was positive in the liver on Kupffer, endothelial, and stellate cells; in the kidney on endothelial and interstitial macrophages and mesangial and nephrocyte base cells; in the lung on alveolar macrophages and endothelial cells; in the spleen on endothelial lymphocytes and activated macrophages, platelets, and megakaryocytes. In the groups with intraperitoneal administration of CN-6, this CD147 marker was also expressed on hepatocytes around the centrilobular venule, on the nephrocyte basement, on a higher number of lymphocytes in the spleen, and on endothelial cells and type I pneumocytes in the lung. Although the IHC profile is positive in these organs, no pathological changes were observed (Figure 15, Figure 16, Figure 17 and Figure 18).

In experimental groups receiving the CN-4 sample by various administration routes, P65 is expressed on endothelial cells in the liver, very little on endothelial cells in the lung, on nephrocyte bases and endothelial cells in the kidney, and on some lymphocytes and lymph nodes. Positive labeling was recorded in organelles from CN-6 with intraperitoneal administration (Figure 15, Figure 16, Figure 17 and Figure 18).

α-SMA positively labelled blood vessel smooth muscle fibers in all organelles examined, some lymphocytes, endothelial cells, and red blood cells in capillaries. No excess pericapillary extracellular matrix was found. Blood vessels were positively marked for red blood cells (Figure 15, Figure 16, Figure 17 and Figure 18).

MHC-II antigens were expressed on some lymphocytes (LB); monocytic lineage cells; a small population of T helper cells; activated T cells; arteriolar, sinusoidal, and venous endothelium; Kupffer cells; spindle cells in the connective tissue of the portal tract; large hepatic veins; and liver capsule with the exception of the liver where duct cells were not labelled with MHCII (Figure 15, Figure 16, Figure 17 and Figure 18). MHC II is expressed on bronchial epithelium, type II pneumocytes, and ciliated epithelial cells.

Cox-2 labeling was noted on cells of the monocytic macrophage system, namely Kupffer cells, splenic macrophages, alveolar macrophages, mesangial cells in kidneys, and at the base of nephrocytes (Figure 15, Figure 16, Figure 17 and Figure 18).

All antibodies taken in the study label endothelial, stellate, and Kupffer cells in the CN-4 exposed batches. The batches exposed to CN-6, intraperitoneal administration also have hepatocytes positively labelled to CD147, p65, αSMA, MHC II, and COX-2. Positive labelling was also recorded in the erythrocytes in capillaries and venules.

The antibodies taken in the study mark endothelial cells, the membrane of the basal pole of nephrocytes, which has multiple invaginations which confers more intense positivity. Cox-2 and MHCII can also distinguish positively labelled mesangial cells. Higher positivity is recorded in the experimental batch CN6 intraperitoneal with both doses.

In all experimental groups, the number of alveolar macrophages detected by Cox-2 increased. Positive and low-positive labelling was also influenced by the positivity of hematomas in septal capillaries and blood vessels.

Low positive labeling CD147, p65, MHC II, and Cox2 was recorded on activated lymphocytes and macrophages in the CN-4 group. Positive MHCII and Cox-2 labelling was recorded in all experimental groups.

NPC has been successfully used for mucosal administration such as oral, nasal, ocular, and pulmonary, due to its mucoadhesive and mucosal permeability properties [19,20]. NPCs have been shown to bind efficiently to intestinal epithelial cells and effectively cross the epithelial barrier to penetrate tissues. This suggests that it is possible that NPCs are taken up by antigen-presenting cells of the mucosal immune system which subsequently transport the delivered antigens to immune initiation sites, such as Peyer’s patches, to induce an immune response [21].

The presence of mononuclear phagocyte system macrophages in the liver, lung, kidney and spleen upon per o.s. administration in our experiment underlines this property of stimulation and distribution by macrophages in the body of chitosan nanoparticles. Following per o.s. and s.c. administration, a significant reactivity was achieved in the lung, followed by the liver, spleen, and kidney. In addition, the i.p. administration induced increased numbers of splenic lymphoid follicles and a slight increase in the number of macrophages in the interstitial space of the urinary tubules in the cortical area of the kidney.

Kupffer cells are specialized in the internalization of foreign nanoparticles, playing an essential role in their uptake, trafficking, and destination in the body (He et al., 2010). After intravenous administration, chitosan nanoparticles are opsonized in the blood before being phagocytosed by macrophages and accumulating in cells of the mononuclear phagocyte system. This passive targeting promotes the accumulation of NPs in the liver [22]. Transport from Kupffer cells to hepatocytes was also studied in mice, and it was observed that the majority of CsNps were localized in hepatocytes after intravenous injection. Micro and macrovesicles in hepatocytes could be given by the accumulation of chitosan nanoparticles in these cells in our experiment.

Kupffer cells internalize NP via several toll-like receptors, mannose, and Fc (Gustafson et al., 2015). The mechanisms involved are macropinocytosis, clathrin-mediated endocytosis, caveolin-mediated endocytosis, and the additional endocytotic pathway [23,24,25]. Clathrin-mediated endocytosis has been shown to be responsible for the internalization of approximately 100–350 nm size, while caveolin-mediated endocytosis is responsible for endocytosis of 20–100 nm size [26,27,28]. Macropinocytosis enables internalization of 0.5–5 μm nanosystems [29].

The liver is a major site of accumulation of CS nanoparticles after intravenous administration [30,31,32]. Administration of CNP by various routes did not produce major histological changes in the organs studied, but there was a proliferation of liver Kupffer cells and alveolar macrophages in the septal space of the pulmonary alveoli without changing its diameter. Intravenous administration of NCs does not cause significant hemodynamic changes, and 30 min after administration of NCs, they accumulate mainly in the liver and lung without causing hemolysis and leukocytosis [33]. The toxicity of CS nanoparticles was manifested by a short-term delay in weight gain as reported in the literature in rats. We did not encounter granulomas in the liver as reported by other researchers [33]. Granulomas found in the lung and liver indicate slow biodegradation of chitosan nanoparticles. Overall, the results obtained indicate good tolerability of intravenous administration of an unmodified chitosan suspension in the dose range studied [33]. The effect of chitosan on the viability of hepatocytes has been investigated in a series of studies on the creation of artificial livers in which chitosan served as a framework (nanofibre scaffold) for hepatocyte cultures. The presence of CS fibres in the intercellular space, performing a supportive function, improved the function of hepatocytes [34,35].

The biocompatibility of NCs is also explained by the fact that, under in vivo conditions, negatively charged plasma proteins are adsorbed on the surfaces of nanoparticles, thus preventing erythrocyte aggregation and hemolysis. The process of thrombosis during systemic NC incorporation is directly dependent on the magnitude of positive NCs’ charge. Thus, pronounced agglutination, hemolysis, and intravascular thrombosis develop in the case of NCs with high charge and high concentration [36], whereas NCs with low charge adsorb coagulation factors, mainly fibrinogen, upon contact with blood and cause only weak inhibition of platelet aggregation [37]. In addition, the charge of NCs and their derivatives depends on the pH of the medium and is determined by the concentration of amino groups in the polymer molecule [38,39]. Resuspension in saline reduces NCs load, thus allowing intravenous administration of those without the risk of hemolysis [38] at particles doses of 4–6 mg/kg [40,41,42]. Including s.c. and i.p. administration was possible in the present experiment without thrombosis or hemolysis phenomena. Lee et al. [43] suggest that the lungs may be the primary barrier organ for intravenously administered CPN. This phenomenon has already been described and has, in fact, been proposed as a targeted delivery strategy for the lungs [43]. Proliferation of alveolar macrophages was also observed in our experiment regardless of the route of administration.

After entering the body, NMs (>6 nm) predominantly accumulate in organs of the mononuclear macrophage system (MPS), such as the liver, spleen, lungs, etc. [40], due to subsequent recognition and internalization by the macrophages of this system [44]. Some particles, such as gold NPs (40 nm) and [45] dextran-coated magnetite NPs (core diameter of 8–10 nm) [46,47] are gradually degraded in cells over several months or even longer periods and sequestered in MPS for a long period of time. Indeed, it has been observed that long-term sequestration in MPS induces some potential side effects, especially immunotoxicity [48]. Accordingly, removal of NCs from MPS organs, especially the liver, sequesters 30–99% of administered NCs from the bloodstream [40] being essential for the clinical safety of NCs. Therefore, it is necessary that all injected NCs are completely removed within a reasonable period of time [49]. Following distribution in hepatocytes, the hepatobiliary–fecal excretion route has been shown to be the primary elimination pathway for CsNps in addition to the reno-urinary excretion pathway. Elimination of CsNps in mice was a lengthy process with a half-life of approximately 2 months.

CD147, a transmembrane glycoprotein with two immunoglobulin-like glycoproteins of the immunoglobulin G (IgG) superfamily, is widely expressed on the surface of various cells, activated lymphocytes, and epithelial cells including cancer cells. As a metalloproteinase inducer, CD147 has been shown to be involved in multiple biological processes such as immune response, tumour progression, and tissue repair. CD147 is recognized as a regulator of lipid metabolism in a variety of cell types and autophagy [50]. In experimental groups, this marker was captured on various cells, but no histopathological changes were reported. Modified CS nanoparticles activate macrophages and are phagocytosed by them [51], processes that lead to aseptic inflammation [38,52]. CD147 is expressed by platelets and is overexpressed following platelet activation [53]. The CS nanoparticles used in this experiment have a small cytotoxic effect and have a weak antiplatelet and anticoagulant effect as mentioned by Sonin et al. [33]. CD147-mediated cell–cell interaction on the cell surface induces platelet activation and via NF-B-factor-mediated monocyte activation will occur [54].

Nuclear factor-κB (NF-κB/p65) is a family of transcription factors that plays a critical role in inflammation, immunity, cell proliferation, differentiation, and survival. Inducible NF-κB activation depends on proteosomal degradation induced by phosphorylation of NF-κB inhibitory proteins (IκB), which retain inactive NF-κB dimers in the cytosol in unstimulated cells. Most of the signalling pathways leading to NF-κB activation converge on the IκB kinase complex (IKK) which is responsible for IκB phosphorylation and is essential for signal transduction to NF-κB. Further regulation of NF-κB activity is achieved by various posttranslational modifications of the basic components of NF-κB signalling pathways. In addition to cytosolic modifications of IKK and IκB proteins, as well as other pathway-specific mediators, transcription factors are themselves extensively modified [55]. The low positive labeling after CN-4 administration demonstrates the presence of modified transcription factors not involved in transcriptional processes. In animals exposed to CN-6, positive labeling may indicate involvement of this factor in transcription. Numerous labelled cells were observed in the spleen of mice from experimental groups. Macrophages are able to integrate an impressive amount of information on the identity and virulence of pathogens, as well as endogenous cues present in their microenvironment, to modulate the immune response for optimal host protection. Central to this capacity are the numerous ways in which NF-κB signaling is modulated based on shifting activation thresholds, integration of information from different classes of pattern recognition receptors, and tight regulation of transcription through rigorous positive and negative feedback loops [54]. The presence of this p65 marker was low positive in the studied organelles with the exception of the CN-6 intraperitoneal exposed batches, denoting a reduced inflammatory response.

Circulating monocytes are recruited into tissues, where they differentiate into macrophages and take part in the process of inflammation or tissue remodeling. According to the traditional concept, macrophages are classified into pro-inflammatory (M1), non-activated (M0), or anti-inflammatory (M2) subsets that play distinct roles in the initiation and resolution of inflammation. More recent experimental findings have led to a substantial update of the monocyte–macrophage nomenclature to include the nature of the polarization signal. In response to proinflammatory stimuli, monocytes can be polarized directly into three subsets of macrophages with M1-type proinflammatory phenotype; interferon-γ-induced macrophages have the strongest proinflammatory properties. When exposed to various anti-inflammatory stimuli, monocytes can differentiate into at least five subsets of M2-type macrophages. Of these, a subset generated upon exposure to IL-4 (IL-13) has the most typical M2-type characteristics. In both humans and mice, differentiation of monocytes into macrophages involves global transcriptome changes that are tightly controlled by various transcriptional regulators and signaling mechanisms [56]. There are many key checkpoints in the transcriptional control and signaling network that trigger either pro-inflammatory or anti-inflammatory polarization [57]. Regulation of NF-κB family transcriptional activity plays a central role in M1–M2 switching and macrophage polarization towards an anti-inflammatory or pro-inflammatory phenotype. The absence of intense p65 labeling also implies insignificant macrophage activation in this experiment.

MHC-II or HLA-DR is a molecule involved in antigen presentation and is the main signal in immunological cooperation [58]. MHC-II antigens are normally expressed on B lymphocytes, monocytic lineage cells, a small population of T helper cells, activated T cells, thymic epithelium, and vascular endothelium [59]. Normal arteriolar, sinusoidal, and central venous endothelium often express MHC II. Kupffer cells have always expressed these antigens. MHC II-positive spindle cell fibroblasts were identified in the connective tissue of the portal tract, large hepatic veins, and liver capsule: most shared antigens common to all leukocytes and reacted with MHC II. Bile duct epithelium expressed MHC II in primary biliary cirrhosis, large duct obstruction, and drug-induced cholestasis, indicating that MHC II positive spindle cells are phenotypically similar to histiocytes. In our experiment liver duct cells were not labeled with MHCII. Although Kupffer cells express lower levels of MHC class II molecules than classical dendritic cells, they are able to interact with T cells. However, unlike dendritic cells, Kupffer cells favor the development of regulatory T cells, thereby promoting immune tolerance [60] which explains the lack of liver injury in this experiment following NC administration. Hepatic macrophages may be highly targeted by nanoparticle drug carriers due to their efficient phagocytosis function in the liver [61].

MHCII is expressed on both bronchial and alveolar epithelium, especially on type II pneumocytes and ciliated cells, and in this experiment as reported in the literature [58].

Various studies have shown that CS and its derivatives can effectively activate antigen-presenting cells and induce cytokine stimulation to produce an effective immune response and promote Th1/Th2 response balance [62]. Catalytically active Cox-2 (and Cox-1) is located in the nuclear envelope (NE) and endoplasmic reticulum (ER), where it mediates PGE2 biosynthesis. Cox-2 dissociated from the nuclear envelope is catalytically inactive [63]. Activated macrophages overexpress these enzymes (Cox-2), which would lead to the production of large amounts of PGs. In addition, NF-κB is a transcription factor that induces copying of proinflammatory genes to produce large amounts of proinflammatory mediators, such as Cox-2, in activated macrophages [55]. The presence of a positive low label at this marker indicates reduced transcription of proinflammatory mediators. Although a large number of alveolar macrophages can be observed in the lung, low positive for alpha SMA indicates a lack of synthesis of collagen precursors. This is also observed in other organs.

Cyclooxygenases are responsible for the synthesis of prostaglandins from arachidonic acid and are present in two isoforms in the kidney, Cox-1 and Cox-2. Cox-1 is involved in the regulation of basic cellular functions, while Cox-2 is a proinflammatory enzyme and is induced by inflammatory stimuli [64]. Cox-2 labeling has been noted in cells of the monocytic macrophage system, namely Kupffer cells, splenic macrophages, alveolar macrophages, and mesangial cells in the kidney. The absence of an inflammatory process denotes that Cox-2 is in an inactive phase.

The adjuvant effect of CS is mainly evaluated from several points of view such as biocompatibility, biodegradability, and cell permeability [65]. Size and surface charge are key characteristics that define how cells interact with and internalize NCs [66]. The affinity of NCs to the cell membrane is related to the cationic component, a characteristic of CS [67].

Upon oral administration an important role in nanoparticle uptake is played by M cells in the intestinal mucosa. The key point for the initiation of the mucosal immune response is antigen uptake, in the case of the NCs based on CS. Many experiments have shown that M cells can carry various macromolecular substances and microorganisms. After M cells in Peyer plates adsorb NCs, they are actively transported to the underlying immune cells, dendritic cells, to stimulate the local immune system or mucosal immunity [68]. Previous research has shown that CS nanoparticles accumulate predominantly in the macrophages of the monocytic macrophage system [69]. Administered systemically, nanoparticles are taken up by Peyer’s plaque cells, pass into the lymph and then into the general blood circulation, and can subsequently be taken up by the liver, kidney, spleen, heart, and other vital organs [70]. CS nanoparticles are also known as immunomodulators. An important role in this case is played by the size and physicochemical characteristics of the particles that can influence the interaction with immune cells to induce the desired therapeutic benefit [65]. Bioactive CS nanoparticles are internalized by macropinocytosis, clathrin-mediated endocytosis, and phagocytosis. They are then transported intracellularly by endosomes, multivesicular bodies, and lysosomes. Proteomics elucidated that chitosan nanoparticles induced an increase in proteins involved in immunoregulatory functions and antioxidant activities. They also promoted the production of anti-inflammatory/pro-regenerative mediators but suppressed pro-inflammatory ones. Therefore, CS nanoparticles could prevent persistent post-treatment inflammation [71].

Most research that has addressed the toxicity of chitosan-based nanoparticles has typically conducted biocompatibility studies 2–4 days after intravenous administration in animal models [72,73,74,75]. However, the toxicity determined from these types of studies, which use short observation periods for long-circulating dispersed solutions [72,74], may not be fully representative because neither biodistribution nor biodegradation processes are completed within these short periods. However, studies over longer observation periods after intravenous administration in subacute or chronic experiments are rare [33], even though long-term observations of biological effects of dispersed systems are no less important than acute toxicity analysis. Based on this consideration, we chose chronic exposure with per o.s., s.c., and i.p. administration.

## 4. Conclusions

Interfacial condensation of CS with poly(NVPAI) is an effective method of obtaining NCs capable of encapsulating, transporting, and delivering drugs. The morphological characteristics of NCs as well as their physicochemical properties can be modeled by varying some parameters of the preparation process (the ratio between the functional groups of the polymers involved in the reaction, the ratio of the phases in which the polymers are solubilized). Consistent with these properties also varies the ability of NCs to encapsulate and release the model drug in a simulated physiological environment (pH = 7.4).

Low positive labeling at all markers taken in the study in experimental groups subjected to CN-4 sample denotes biocompatibility and does not result in histological changes. These NCs have demonstrated the ability to stimulate cells of the mononuclear phagocyte system in the liver, lung, spleen, and kidney without producing inflammatory changes. When administered per o.s., proliferation of alveolar macrophages was observed, followed by Kupffer cells. Administration of CN4 and CN6 i.p. at a dose of 0.02 mL induced micro- and macrovacuoles of hepatocytes. Overexpression of CD147, p65, MHCII, and Cox-2 markers was observed in the macrophages of the organs studied in all experimental groups. The absence of alpha SMA labelling denotes the absence of an inflammatory process.

From this study, it can be concluded that the obtained NCs have the advantage of being biocompatible, non-toxic, and safe for in vivo administration as drug delivery systems.

## Figures and Tables

**Figure 1 polymers-14-01811-f001:**
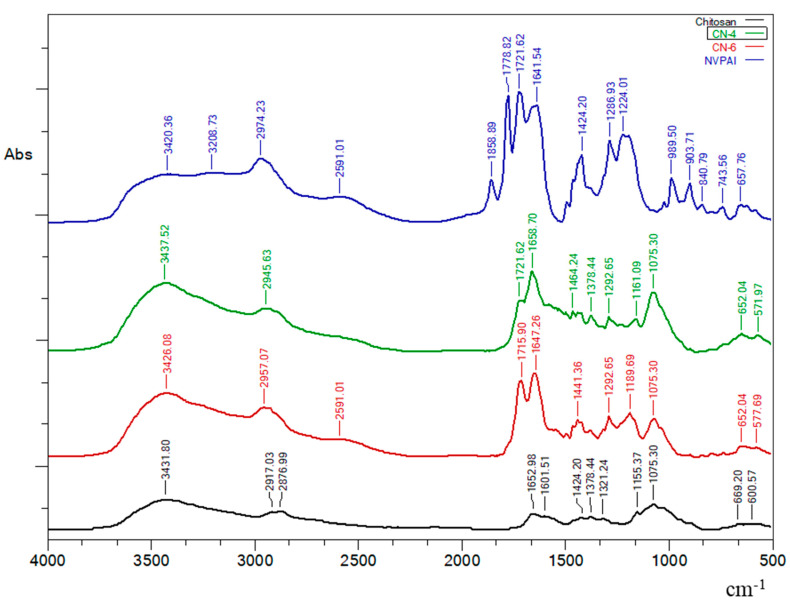
FTIR spectra of CS, NVPAI, CN-4 and CN-6 samples.

**Figure 2 polymers-14-01811-f002:**
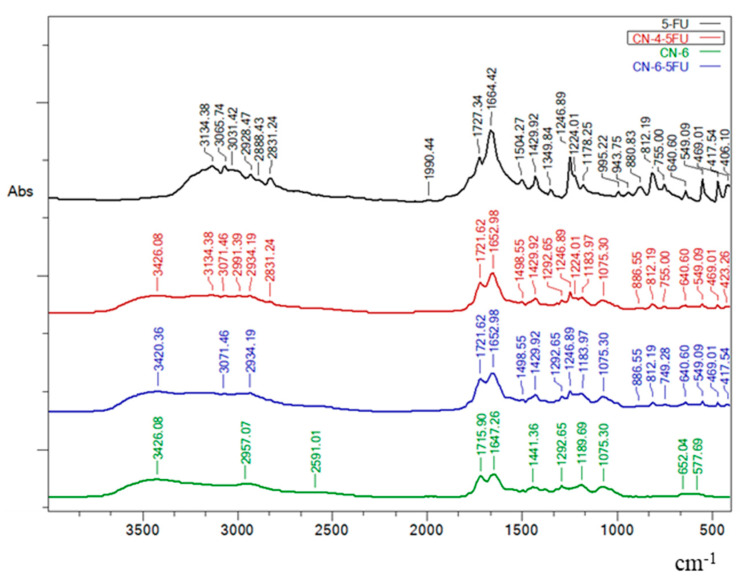
FTIR spectra of simple 5-FU, CN-6 without drug, and drug-loaded NCs (CN-4-5FU and CN-6-5FU).

**Figure 3 polymers-14-01811-f003:**
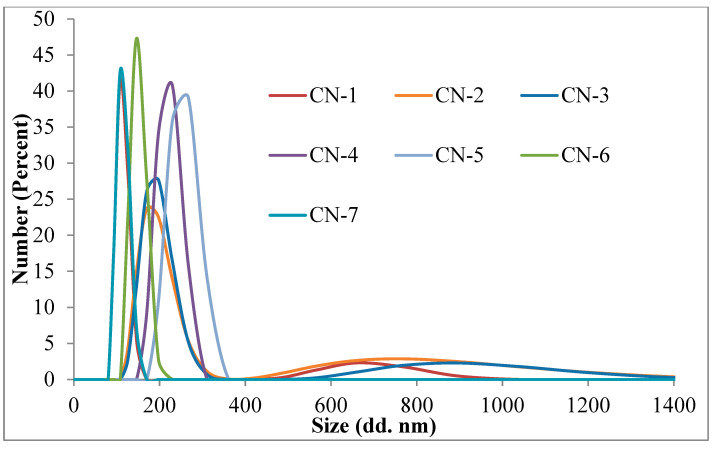
Size distribution curves of NCs in acetone at room temperature.

**Figure 4 polymers-14-01811-f004:**
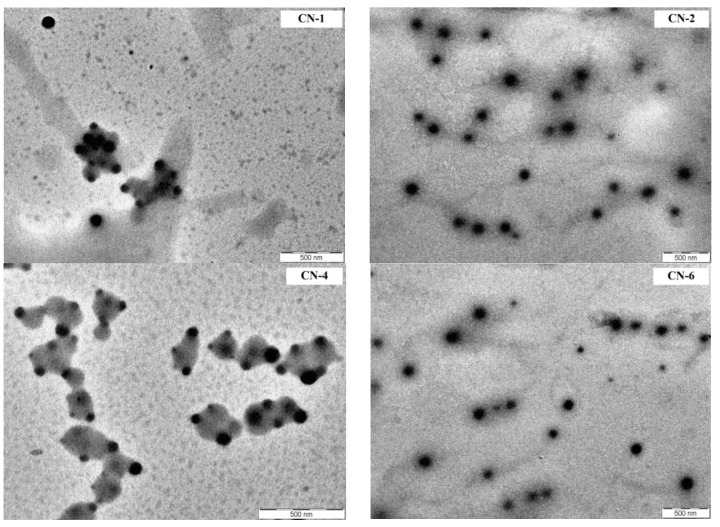
TEM images of CN-1, CN-2, CN-4 and CN-6 samples.

**Figure 5 polymers-14-01811-f005:**
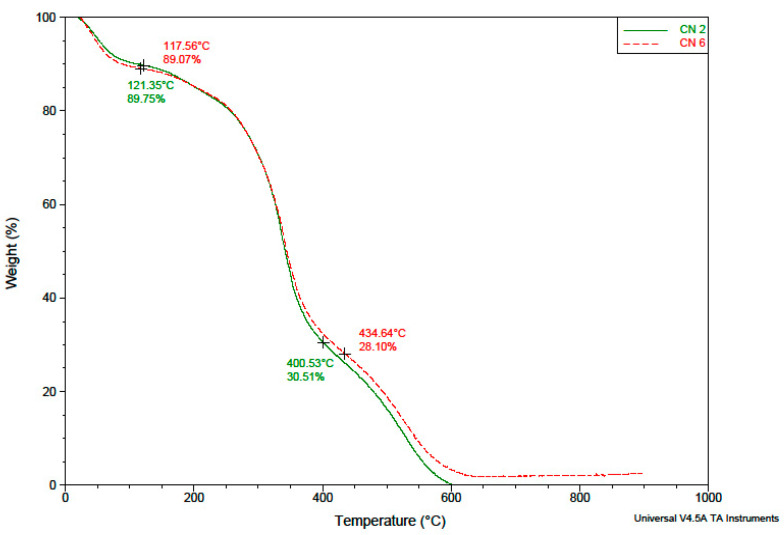
TGA chromatograms of NCs samples.

**Figure 6 polymers-14-01811-f006:**
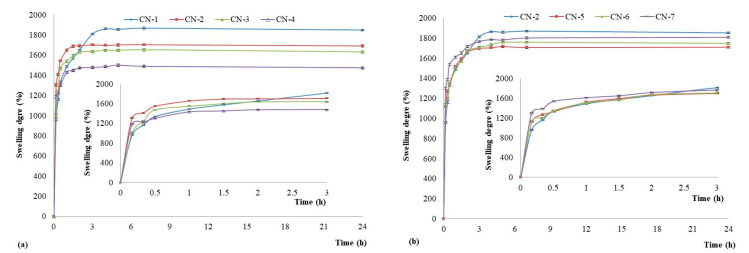
The swelling kinetics curves of the NCs in alkaline conditions (pH = 7.4) for samples (**a**) CN-1, CN-2, CN-3, and CN-4; (**b**) CN-2, CN-5, CN-6, and CN-7.

**Figure 7 polymers-14-01811-f007:**
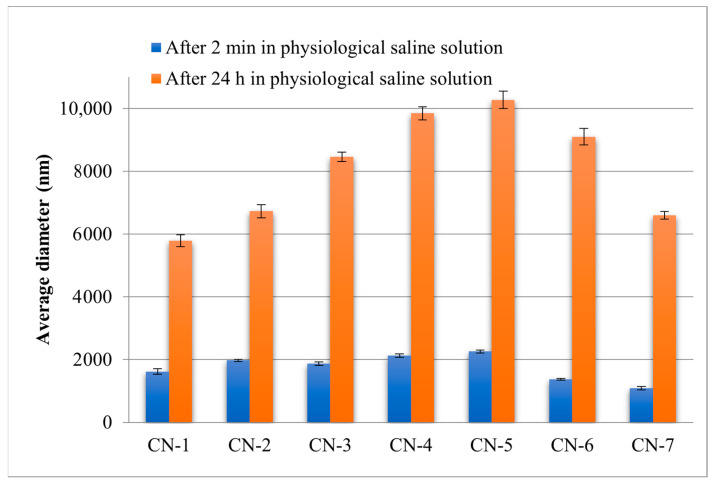
Evaluation of the average diameter of the NCs after 2 min and 24 h in physiological saline solutions.

**Figure 8 polymers-14-01811-f008:**
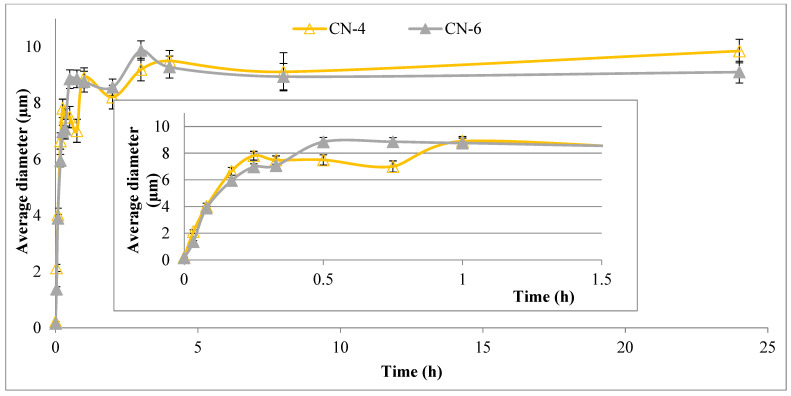
The influence of the aqueous environment on the size of CN-4 and CN-6 samples over time.

**Figure 9 polymers-14-01811-f009:**
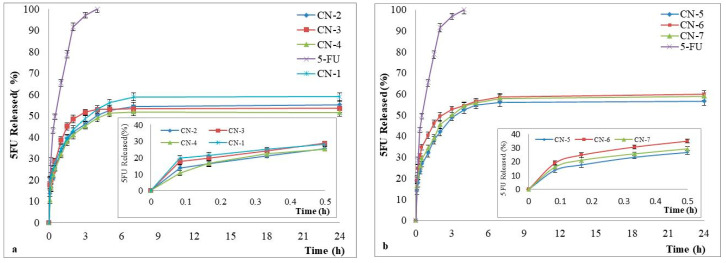
In vitro release kinetics of 5-FU from NCs in phosphate buffer solution (pH 7.4), with a zoom insert of the release kinetics between 0 and 30 min, for samples (**a**) CN-2, CN-3, and CN-4; (**b**) CN-5, CNM-6, and CN-7.

**Figure 10 polymers-14-01811-f010:**
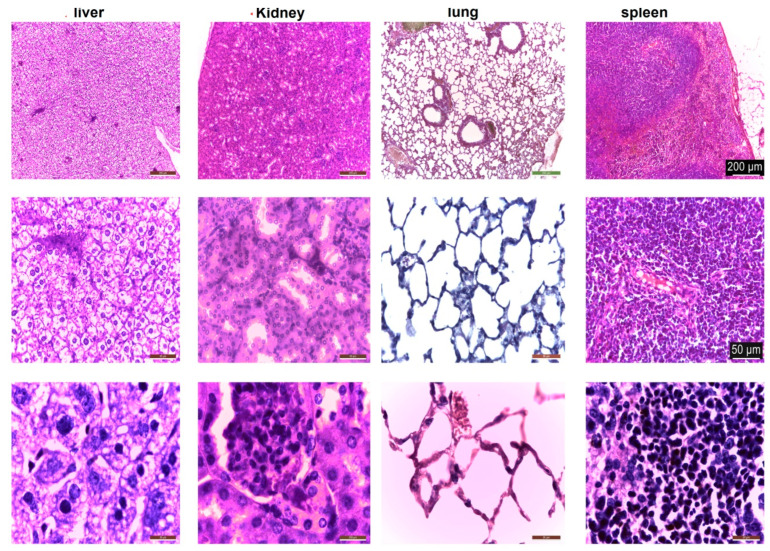
Histological structure of liver, kidney, lung, and spleen in control mice. HE staining.

**Figure 11 polymers-14-01811-f011:**
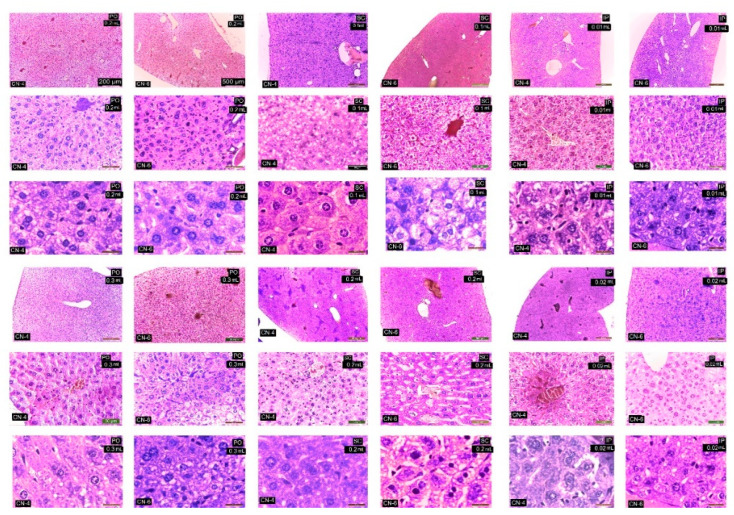
Histological structure of the liver in experimental groups. HE staining.

**Figure 12 polymers-14-01811-f012:**
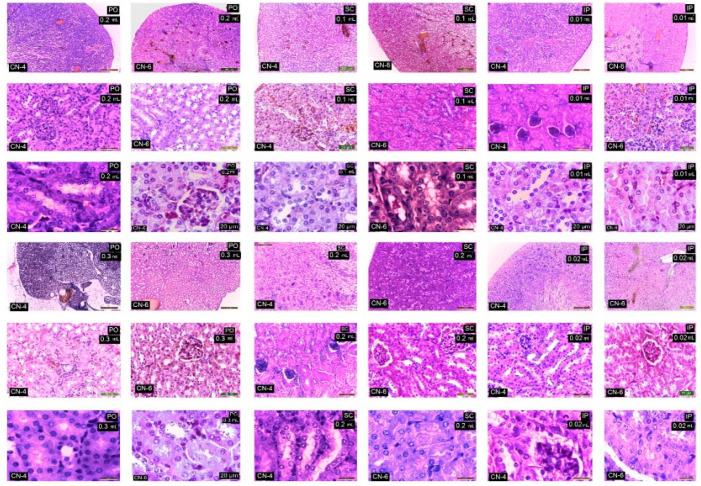
Kidney in experimental groups. HE stain.

**Figure 13 polymers-14-01811-f013:**
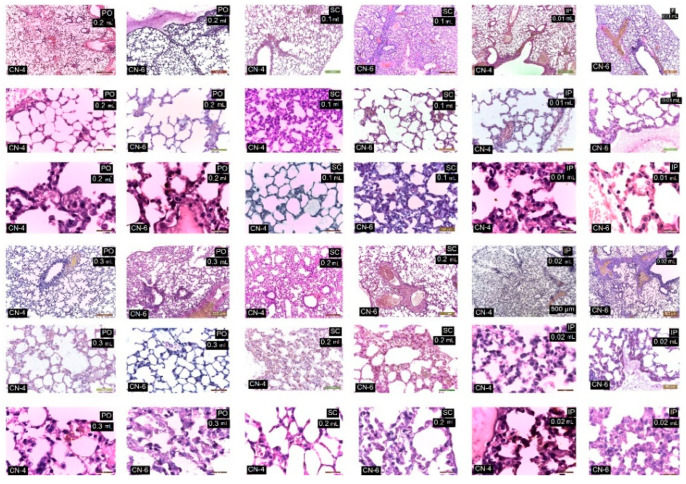
Lungs in the experimental group. HE staining.

**Figure 14 polymers-14-01811-f014:**
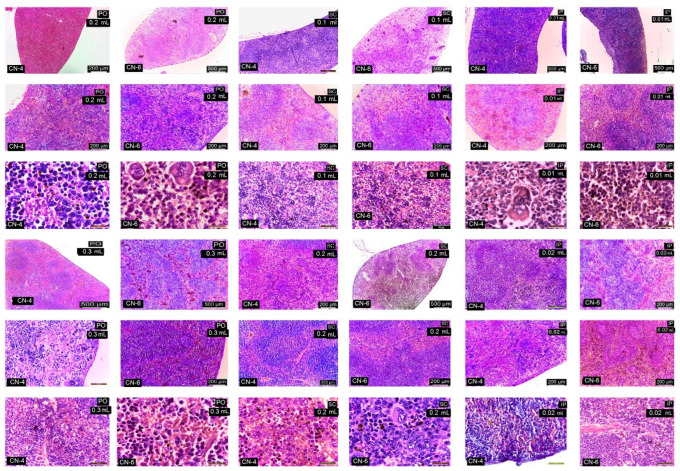
Spleen in the experimental group. HE staining.

**Figure 15 polymers-14-01811-f015:**
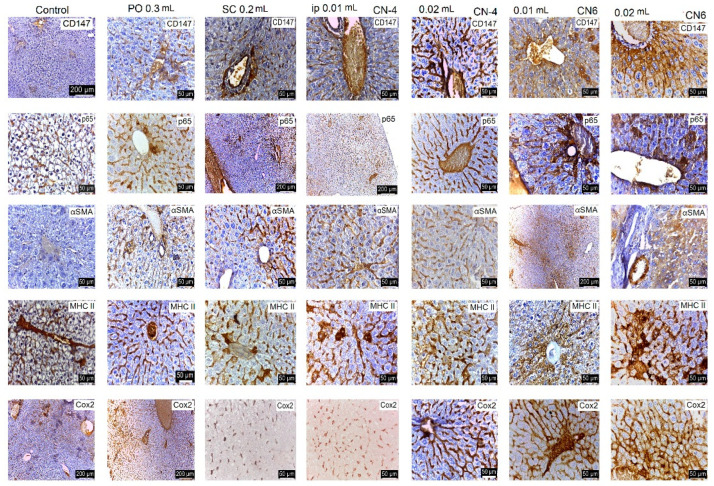
IHC marking in the liver.

**Figure 16 polymers-14-01811-f016:**
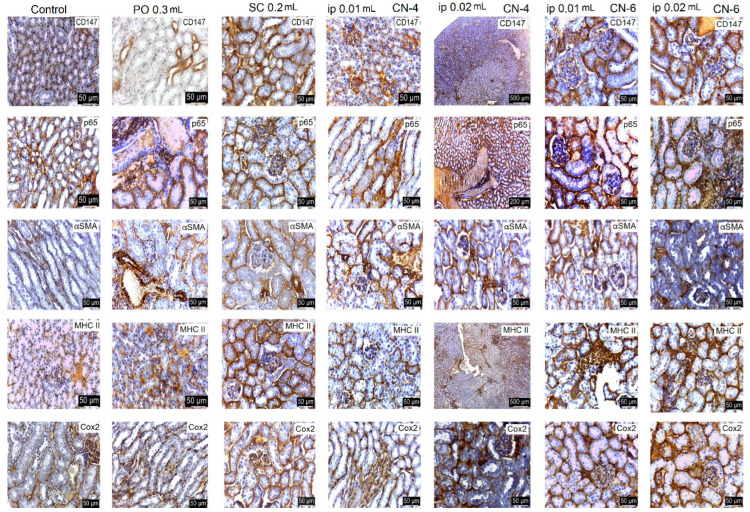
IHC labeling in kidneys.

**Figure 17 polymers-14-01811-f017:**
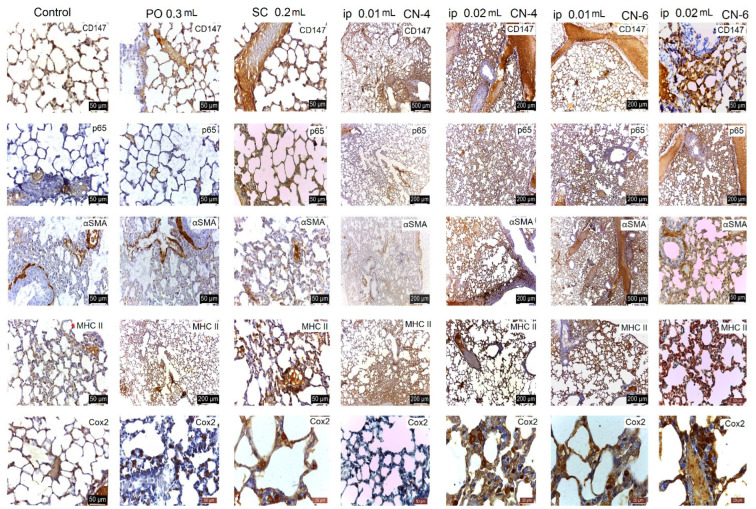
IHC labelling in the lung.

**Figure 18 polymers-14-01811-f018:**
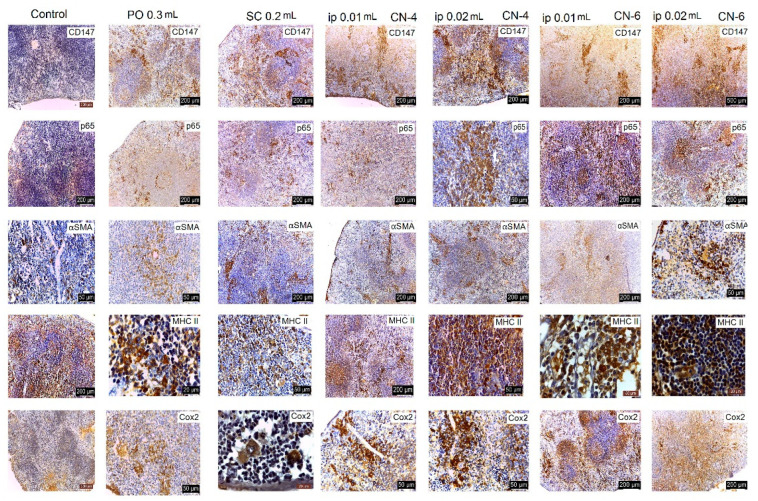
IHC labeling in the spleen.

**Table 1 polymers-14-01811-t001:** Experimental parameters used for the preparation of NCs.

Sample Code	CS/Poly(NVPAI) (mol/mol)	% CS (*w*/*v*)	Aqueous Phase/Organic Phase Ratio (*v*/*v*)
CN-1	0.2/1	0.50	1:2.0
CN-2	0.3/1	0.75	
CN-3	0.4/1	1.00	
CN-4	0.5/1	1.25	
CN-5	0.3/1	0.75	1:2.5
CN-6			1:3.0
CN-7			1:3.5

**Table 2 polymers-14-01811-t002:** Yield data for the obtained NCs.

Sample Code	CN-1	CN-2	CN-3	CN-4	CN-5	CN-6	CN-7
Yield (%)	28	39	45	49	62	77	83

**Table 3 polymers-14-01811-t003:** Diameter in volume, polydispersity index (PDI), and Zeta potential values of NCs samples.

Samples Code	Dv (nm) in Acetone	PDI	Dv (nm) in Physiological Saline Solution	PDI	ZP (mV) in PBS (pH = 7.4)
CN-1	107 ± 0.11	0.96 ± 0.04	1621 ± 84.42	0.43 ± 0.12	−19.8 ± 0.01
CN-2	188 ± 0.23	0.73 ± 0.02	1976 ± 34.51	0.42 ± 0.09	−18.4 ± 0.04
CN-3	192 ± 0.51	0.83 ± 0.04	1869 ± 57.84	0.42 ± 0.04	−18.0 ± 0.05
CN-4	220 ± 0.53	0.64 ± 0.03	2131 ± 51.61	0.41 ± 0.09	−16.8 ± 0.01
CN-5	250 ± 0.30	0.94 ± 0.03	2256 ± 43.82	0.31 ± 0.09	−18.4 ± 0.04
CN-6	150 ± 0.22	0.68 ± 0.06	1371 ± 30.91	0.23 ± 0.02	−8.5 ± 0.02
CN-7	114 ± 0.21	0.94 ± 0.04	1089 ± 53.84	0.46 ± 0.03	−11.06 ± 0.07
CN-1-5FU	169 ± 0.73	1.22 ± 0.07	2263 ± 157.62	1.30 ± 0.24	−20.07 ± 0.24
CN-2-5FU	226 ± 0.63	1.02 ± 0.10	2156 ± 203.41	1.27 ± 0.39	−20.57 ± 0.38
CN-3-5FU	254 ± 0.42	1.04 ± 0.04	2396 ± 107.04	1.07 ± 0.68	−21.1 ± 0.32
CN-4-5FU	240 ± 0.87	0.92 ± 0.08	2421 ± 124.95	1.12 ± 0.60	−21.52 ± 0.17
CN-5-5FU	297 ± 0.43	1.00 ± 0.12	2947 ± 191.39	1.10 ± 0.35	−22.22 ± 0.04
CN-6-5FU	227 ± 0.44	0.89 ± 0.05	1887 ± 54.22	0.54 ± 0.23	−23.21 ± 0.34
CN-7-5FU	181 ± 0.69	0.93 ± 0.24	1594 ± 79.73	0.81 ± 0.16	−22.8 ± 0.18

**Table 4 polymers-14-01811-t004:** Drug encapsulation efficiency values.

Sample Code	Encapsulated Drug g/g NCs	The Encapsulation Efficiency (%)
CN-1	0.317 ± 0.001	42.3 ± 0.173
CN-2	0.244 ± 0.003	32.6 ± 0.333
CN-3	0.219 ± 0.001	29.3 ± 0.120
CN-4	0.198 ± 0.002	26.5 ± 0.289
CN-5	0.249 ± 0.001	33.2 ± 0.120
CN-6	0.258 ± 0.006	34.5 ± 0.866
CN-7	0.266 ± 0.004	36.0 ± 0.577

**Table 5 polymers-14-01811-t005:** Assessment of cellular and vascular changes produced in the organs under study.

Organ/Lesions	Control	PO 0.3 mL	SC 0.2 mL	Ip 0.01 mL CN-4	Ip 0.02 mL CN-4	Ip 0.01 mL CN-6	Ip 0.02 mL CN-6
Liver							
-cords are radially arranged from the centrilobular venule to the periphery of the lobule	+	+	+	+	+	+	+
No of Kupffer cells/12,879.8 mm^2^	3.5	6–7	6–7.5	7–8	8–9.3	9–10.5	9–11
Vascular congestion	−	−	−	−	++	++	++
**Kidney**							
Malpighian corpuscles and urinary tubules show changes	−	1	1	1	1	3	3
No of. Interstitial macrophages/12,879.8 mm^2^	−	2–4	2–5	4–6	4–6.6	5–6	5–7.5
Increased in volume of nephrocytes of the proximal convoluted tubules	−	−	−	+	+	++	++
Vascular congestion	−	−	−	+	+	+	++
**Lungs**							
thin-walled alveoli	+	+	+	+	+	+	+
No of alveolar macrophages/12,879.8 mm^2^	3–4	14–22	14–24	9–11	9–13	9.5–17	9–18.5
Vascular congestion	−	−	−	+	+	++	++
**Spleen**							
Lymphoid follicles and lymphatic cords	−	−	++	++	++	++	++
Vascular congestion	−	−	+	+	+	++	++
Megakaryocytes and pigment cells	−	−	+	+	+	++	++

Scoring of HE histological lesions was done by assessing changes and scoring as follows: no change (−), minor (+), medium (++).

## Data Availability

The data presented in this study are available on request from the corresponding author.

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
