# Peer review of "Assessment of Physicochemical and In Vivo Biological Properties of Polymeric Nanocapsules Based on Chitosan and Poly(N-vinyl pyrrolidone-alt-itaconic anhydride)"

_polymers, 2022, doi:10.3390/polym14091811_

Round 1

Reviewer 1 Report

The manuscript entitled "Assessment of physicochemical and in vivo biological properties of polymeric nanocapsules based on chitosan and poly (Nvinyl pyrrolidone-alt-itaconic anhydride)" is well written, commented and organized. The results can be easily tracked.
The manuscript can be accepted for publication after a minor revision:
- Adding the statistical analysis of the presented data

Reviewer 2 Report

In this article, Dellali et al. developed polymeric nanocapsules made of chitosan and poly(NVPAI). The authors assessed the nanoparticles' physicochemical properties in vitro and evaluated their in vivo safety. I have important concerns about some of the results presented, and the lack of discussion. Numerous issues need to be addressed before publication, including repeating assays.

Major revisions:

- At the end of the introduction, the authors refer to a hydrophobic model drug. The author should mention which drug was chosen and why.

- One of my biggest concerns is why the authors determined the size and zeta potential of the nanoparticles in anhydrous acetone? The authors state that it is to avoid the swelling of the particles (Pag. 4), but obviously, the nanoparticles were not administrated to the animals in acetone. The nanoparticles were administrated to the mice in a saline solution. Therefore, the authors should assess the physicochemical properties of the particles using the same medium.

- Subsection 2.3.3: The authors should give more details on samples preparation.

- Subsection 2.3.5: Please indicate the weight of nanoparticles per mice weight.

- Subsection 2.3.9: How was the drug encapsulated into preformed nanoparticles? Please provide more details and references that support the concept.

- The physicochemical characterization of 5-FU-loaded nanocapsules is missing.

- The statistical analysis is missing. Please perform the statistical analysis of all data, present all the p-values, and discuss the results accordingly.

- Section 3: "This behaviour is normal". The authors should discuss why this is normal and provide references that support the discussion.

- Page 7: Why did the authors only perform FTIR and TEM of 2 formulations (4 and 6)? And then, the thermal behavior was assessed only for CN-2 and CN-6.

- Figure 2: Size distribution curves of CN-5 and CN-7 are missing.

- Page 8: The physicochemical characterization of CN-7 is missing.

- Table 2: Please add the standard deviation of PDI values.

- Table 2: Please pay attention to how the results (mean ± SD) are presented, particularly concerning the number of decimal places.

- Figure 3: The authors should provide images of better quality.

- Subsection 3.2.: The authors did not discuss the impact of the aqueous/organic phase ratio on the nanocapsules' physicochemical characterization.

- Subsection 3.3.: The authors aim with the thermal behavior determination two things: 1) “to have additional evidence that their membrane is made of both polymers” and 2) “to determine whether it is possible to heat sterilize them before administration”. But none of the questions were addressed or discussed.

- Subsection 3.3.: "Obviously, the composition of the NCs shell does not qualitatively influence its thermal degradation behavior." The authors only present the TGA chromatogram of 2 formulations (CN-2 and CN-6). Why?

- Figure 5: Why does the swelling degree start at 1000%?

- Subsection 3.4.: “The size, the composition and the preparation parameters of NCs have an important influence on the swelling properties.” And have the swelling phenomena effect on NCs size? This should be evaluated over time.

- Page 10: “An identical tendency was observed with the increase of CS amount (Fig.5.b).” Figure 5b refers to the aqueous/organic phase ratio, not the CS amount.

- How about the survival curve and bodyweight of mice through the experiment?

- Why did the authors only test two formulations in vivo (CN-4 and CN-6)?

- Subsection 3.5.: The authors did not attempt to discuss the significance of the observed changes. Proof of this is that no references were used between pages 10 and 18. This section must be worked intensively.

- IHC labelling: PO 0.2 mL and SC 0.1 mL are missing.

- I suggest moving sections 3.6. and 3.7. before the in vivo result to present all the in vitro data first, and then the in vivo.

Table 3: CN-1 formulation is missing.

Table 3: The results should be presented as mean ± standard deviation.

Figure 15: The release kinetic of the free drug is missing.

Figure 15: The release kinetic of CN-1 is missing.

Minor revisions:

- The words "nanocapsules", "chitosan" and "poly(N-vinyl pyrrolidone-alt-itaconic anhydride)" already appear in the title. I suggest removing or replacing these words with other keywords to increase the article's visibility. Some suggestions: Nanoparticles, polymers, safety,…

- In the introduction, the authors said that SEM was used, but the methodology refers to TEM.

- Subsection 2.2 Please replace Cs with CS.

- Table 1: The yield data should be presented in the results section instead of the methodology.

- Subsection 2.3.5: "Suspensions of CN-4 and CN-6 were obtained by adding 5 ml of saline over 6.25 mg powder, the suspension being prepared approximately 2 hours before use." This sentence should appear before the previous.

- Subsection 2.3.5: "Suspensions of CN-4 and CN-6 were obtained by adding 5 ml of saline over 6.25 mg powder". Please specify the 'saline' solution used.

- Subsection 2.3.5: How were mice euthanized?

- Subsection 2.3.5: "Organ samples were fixed with 10% formalin solution for 24 hours, dehydrated with ethyl alcohol, clarified with xylene, paraffin embedded, sectioned at 4µm, HE and IHC stained with CD147, p65, ⍺-SMA, MHC II, , Cox2 markers." I suggest removing this sentence as the content appears in the following subsection.

- Subsection 2.3.9: "Briefly, 20 mg NCs were dispersed in 1.5 ml aqueous drug solution with a concentration of 10 mg/ml 5-FU". Which aqueous solution? Ultrapure water? A buffer? Please clarify.

- Subsection 2.3.1.: "the amount of drug released from magnetic NCs". Magnetic?

- Figure 5 and Figure 15: Please provide time in hours for a better understanding.

- Subsection 3.5.: “For the in vivo testing, first of all was tested a control group and then experimental groups which received the NCs”. I suggest removing this sentence and providing a better introduction.

- Subsection 3.6.: The abbreviation of 5-Fluorouracil was already introduced.

- Table 3: Please use ‘Encapsulation efficiency’ instead of ‘The loading efficiency’.

Reviewer 3 Report

Congratulations! The article is very well organized. The results are of high originality and scientific impact. It was a pleasure to read it.

I have a suggestion to the In vivo subchapter to compare your work to similar one or to mention if your study is pilot.

Reviewer 4 Report

Comments: The work is an interesting manuscript,  However there are few conceptual errors should to be addressed before accepting manuscript for publication.

  • Marker name of IHC should please to be written down for each figure
  • The biological observations should be summarized in table for all kinds of administrations and then comparison among them should be provided.
  • The manuscript is missing comprehensive discussion concerning cytotoxicity of Chitosan alone, Poly(NVPAI) alone or The mixture . Cytotoxicity of such these materials should have been studied.
  • Quantification analysis for each IHC marker was not obtained causing weakness point in evaluating the significant value of the results
  • From such this study, authors should to obtain biological factors (parameters) that will be used  later to control application of NPs.
  • Authors should to clarify that all organic solvents used in fabrication, were exactly removed well and NPs were washed and purified before animal experiments.
  • Cytokines and Interleukin were not studied.
  • It is written in conclusion that mixture was safe and non-toxic. However, IHC provides clear expression of many used markers.
  • Many references are strongly suggested to fight this manuscript

Doi: 10.1016/j.ijbiomac.2021.04.131.

Doi: 10.1016/j.ijbiomac.2021.12.073.

Round 2

Reviewer 2 Report

The authors have made a lot of efforts to address all my suggestions. However, the new data related to the characterization of the nanoparticles in PBS clearly reveals aggregated populations. PDI values of 1 indicate a highly polydisperse sample with multiple particle size populations. Furthermore, the size of the particles is incredibly high, with sizes near 2 µM when the drug is loaded. Since the system is intended for biomedical applications, microparticles are not desired.

Some suggestions:

Figure 4: The authors should provide images of better quality.

Figure 9: The authors should remove the old figure.

Table 4: Please use ‘Encapsulation efficiency’ instead of ‘The loading efficiency’.

Reviewer 4 Report

Although comments were revised by authors point by point. However, there is still unclear discussion was seen in histology profiles. 

  1. There is clear Eosinophilic Cytoplasmic structure  (bright-pink dye), seen in groups treated by  CN-4 (0.2) Fig. 11 and in CN-6, Fig.12
  2. There is foamy cytoplasm was clear seen in groups treated by CN-4 and CN-6 (0.1) Fig. 11
  3. There is necrosis in groups treated by CN-6 (0.02), Fig. 11 Finally,  it is strongly  suggested   to  revise the histopathology again

Round 3

Reviewer 2 Report

It is widely recognized that highly polydisperse samples are not suitable for biomedical applications since the presence of a polydisperse suspension (the sample has a very broad particle size distribution) can lead to unexpected variations in the particles’ behavior.

Reviewer 4 Report

Authors have revised the comments step by step. Manuscript is more acceptable.